# Characterization of Novel and Known Activators of Cannabinoid Receptor Subtype 2 Reveals Mixed Pharmacology That Differentiates Mycophenolate Mofetil and GW-842,166X from MDA7

**DOI:** 10.3390/ijms26104956

**Published:** 2025-05-21

**Authors:** Alice L. Rodriguez, Aidong Qi, Allie Han, Haley E. Kling, Marc C. Quitalig, Aaron M. Bender, Lisa Barbaro, David Whomble, Craig W. Lindsley, Colleen M. Niswender

**Affiliations:** 1Warren Center for Neuroscience Drug Discovery, Vanderbilt University, Nashville, TN 37232, USA; alice.rodriguez@vanderbilt.edu (A.L.R.); aidong.qi@vanderbilt.edu (A.Q.); xueqing.han@vanderbilt.edu (A.H.); haleyekling@gmail.com (H.E.K.); marc.quitalig@vumc.org (M.C.Q.); aaron.bender@vanderbilt.edu (A.M.B.); lisa.barbaro@vanderbilt.edu (L.B.); david.l.whomble@vanderbilt.edu (D.W.); craig.lindsley@vanderbilt.edu (C.W.L.); 2Department of Pharmacology, Vanderbilt University School of Medicine, Nashville, TN 37232, USA; 3Department of Chemistry, Vanderbilt University, Nashville, TN 37232, USA; 4Vanderbilt Brain Institute, Vanderbilt University, Nashville, TN 37232, USA; 5Vanderbilt Kennedy Center, Vanderbilt University Medical Center, Nashville, TN 37232, USA; 6Vanderbilt Institute of Chemical Biology, Vanderbilt University, Nashville, TN 37232, USA

**Keywords:** cannabinoid receptor, allosteric modulator, mycophenolate mofetil, GW-842,166X, MDA7, brain disorders

## Abstract

CB_1_ and CB_2_ cannabinoid receptors are members of the GPCR superfamily that modulate the effects of endocannabinoids. CB_1_ is the most abundant CB receptor in the central nervous system, while CB_2_ is present both peripherally and in the brain. CB_2_ plays a role in inflammation, as well as neurodegenerative and psychiatric disorders. To identify new ligands for CB_2_, we screened a library of FDA-approved drugs for activity at the receptor using a thallium flux assay, resulting in the discovery of the immunosuppressant mycophenolate mofetil as a potent, selective activator of CB_2_. Further characterization of the compound confirmed agonist activity in a variety of complementary assays, including PI hydrolysis, cAMP inhibition, and β-arrestin recruitment. Radioligand binding assays established a non-competitive interaction with the site occupied by [^3^H]CP55,940. CB_2_ agonists GW-842,166X and MDA7 were also profiled, revealing that GW-842,166X exhibits a similar activity profile to mycophenolate mofetil, whereas MDA7 presents a distinct profile. These differences provide insight into the complex CB_2_ pharmacology impacting preclinical and clinical studies, and ultimately, new treatment strategies for brain disorders.

## 1. Introduction

The endocannabinoid system is comprised of two primary receptor subtypes, the cannabinoid receptor subtype 1 (CB_1_) and subtype 2 (CB_2_). While CB_1_ is more abundantly expressed in the brain, CB_2_ is expressed in the immune system and was initially thought to be a peripheral receptor [1,2]. CB_1_ is a potential therapeutic target for indications including nausea [3], psychosis [4], anxiety [5], and pain [6]. Subsequent studies characterized CB_2_ expression in the brain [7,8,9,10], and the CB_2_ receptor has emerged as a therapeutic target for brain disorders ranging from anxiety [11] and depression [12] to addiction [13] and schizophrenia [14,15].

A number of CB_2_ receptor agonists have been developed and tested in clinical trials for a variety of indications, including pain and Alzheimer’s disease, and several compounds have been shown to be safe for human administration. GW-842,166X is a CB_2_ selective agonist developed by GlaxoSmithKline [16,17] and was evaluated in Phase I trials for the indications of pain and inflammation (NCT00511524) and in Phase II trials for dental pain (NCT00444769). The trials were either withdrawn or completed with no results posted. MDA7 (NTRX-07) is a CB_2_-preferring agonist developed by Naguib et al. and NeuroTherapia [18,19,20]; it progressed into Phase I trials for the indications of neuropathic pain (NCT04375436) and Alzheimer’s disease (NCT06194552), and according to press releases, this compound is progressing into Phase II studies for Alzheimer’s disease.

As an alternative to direct CB_2_ activation, the use of allosteric modulators may provide an improved approach by targeting neurotransmitter-dependent activity [21]. The spatial and temporal control made possible with allosteric modulators, along with improved subtype-selectivity, suggests that efficacy may be improved with fewer adverse side effects [22,23]. The first published small-molecule positive allosteric modulator (PAM) of CB_2_, EC21a, has been shown to have diverse functional activity [24,25]. While reported initially as a potentiator of CB_2_, it has subsequently shown inverse agonist activity in a variety of independent assays; thus, a pure PAM has yet to be identified [26,27]. To discover novel compounds with CB_2_ PAM activity, we screened a collection of approximately 1000 FDA-approved drugs and identified mycophenolate mofetil as a putative PAM of the CB_2_ receptor. Additional profiling, however, revealed that this compound was actually an agonist of CB_2_. Mycophenolate mofetil is a widely used immunosuppressive agent [28,29,30]; this compound demonstrated CB_2_ activity in a variety of assays, including G Protein Inwardly Rectifying Potassium (GIRK) channel-mediated thallium flux, phosphoinositide (PI) hydrolysis, cAMP inhibition, and β-arrestin recruitment. It was also selective for CB_2_ versus CB_1_. In contrast to the parent molecule, the active form of the drug that is responsible for immunosuppression, mycophenolic acid [28,29,30], was inactive at the CB_2_ receptor. Radioligand binding studies indicate that mycophenolate mofetil does not interact with the binding site occupied by [^3^H]CP55,940 in a competitive manner. During the course of our studies, we also profiled the known CB_2_ agonists, GW-842,166X and MDA7. In receptor-activated GIRK assays, our studies indicated comparable agonist activity between GW-842,166X and mycophenolate mofetil but MDA7 exhibited inverse agonist activity at CB_2_; however, all three compounds demonstrated agonist activity in multiple signaling pathways using cAMP inhibition and β-arrestin assays. With respect to radioligand binding, GW-842,166X exhibited comparable binding activity to mycophenolate mofetil, while MDA7 displayed a different binding profile. The FDA-approved immunosuppressant mycophenolate mofetil represents a novel CB_2_ agonist with unique properties compared to known CB_2_ agonists. While subtle, these differences may provide insight into the mechanism of action of CB_2_ modulators and prove to be significant when evaluating novel compounds as possible therapeutics for brain disorders.

## 2. Results

### 2.1. Identification of Novel Modulators of the CB_2_ Receptor

To identify new compounds capable of potentiating CB_2_ responses to the endogenous agonist 2-AG, we used a thallium flux assay measuring the ability of ligands to enhance 2-AG-induced responses in cells expressing rCB_2_ and GIRK1/2 channels to screen a collection of FDA-approved drugs. Compounds were added to cells loaded with thallium-sensitive dye and allowed to incubate for 2.3 min, whereupon a submaximal concentration (EC_20_) of 2-AG was added to the cells and the signal was monitored for an additional 2.7 min. The ~1000 compound collection was initially screened at a single point using a 10 µM concentration of the compound. Test compounds that potentiated the cellular response to 2-AG by a minimum of 3 standard deviations above the average EC_20_ agonist response were selected as potential PAM hits. Sample kinetic traces of a hit from the screen, compared to EC_20_ and EC_Max_ responses to 2-AG, are shown in Figure 1. A total of 63 putative PAMs were identified; of these, 13 were confirmed as active with concentration-dependent activity at the rCB_2_ receptor.

### 2.2. Mycophenolate Mofetil Is a Potent and Selective Activator of rCB_2_

One hit in particular, mycophenolate mofetil (Figure 2A, white circles), exhibited robust potency and efficacy at rCB_2_ (pEC_50_ = 6.64 ± 0.06, 228 nM; 74 ± 2% Max) and was selected for additional studies. Mycophenolate mofetil is an immunosuppressant used to prevent organ transplant rejection and to treat autoimmune disorders such as lupus [28,29,30]. To determine if mycophenolate mofetil exerts its activity in the thallium flux assay through rCB_2_, we tested it for activity in HEK cells expressing GIRK channels but not rCB_2_. Submaximal concentrations of acetylcholine were used as the agonist, targeting an M_4_ muscarinic receptor expressed in these HEK/GIRK cells [31]. Mycophenolate mofetil was found to be inactive under these conditions (Figure 2A, red circles), suggesting that its activity is dependent on the presence of the rCB_2_ receptor. It was also critical to determine if mycophenolate mofetil was selective for the rCB_2_ versus rCB_1_ receptors, as activation of rCB_1_ receptors can have negative side effects, including anxiety, paranoia, and psychosis [32,33,34]. To determine subtype selectivity, mycophenolate mofetil was tested for activity in cells co-expressing rCB_1_ and GIRK channels using thallium flux. The compound did not enhance the agonist-induced response; however, mycophenolate mofetil exhibited a small inhibition of the rCB_1_ response to 2-AG at high concentrations. The results are shown in Figure 2A (blue circles, pEC_50_ < 5, −5.5 ± 1.1% Max). Mycophenolate mofetil was also tested for activity at the human CB_2_ receptor and found to be active (pEC_50_ = 5.70 ± 0.11; 36 ± 4% Max).

### 2.3. Mycophenolic Acid Is Inactive at CB Receptors

Mycophenolic acid is the active form of mycophenolate mofetil when used as an immunosuppressive drug or for the treatment of autoimmune disorders [35]. The morpholino ethyl ester of mycophenolic acid is a prodrug form that is rapidly metabolized to the active acid in the liver. Mycophenolic acid is a potent inhibitor of inosine-5′-monophosphate dehydrogenase [36]. To test the activity of this active metabolite, the acid was evaluated on rCB_2_, rCB_1_, and parental HEK/GIRK cells using the thallium flux assay. It was found to be inactive when tested in each of these cell lines (Figure 2B), indicating that mycophenolate mofetil, as opposed to the acid, is the active form of the molecule in cells expressing rCB_2_.

### 2.4. Mycophenolate Mofetil Has Similar Activity in the Presence and Absence of 2-AG in Thallium Flux Assays

The initial high-throughput screen and secondary assays were performed in the presence of a submaximal (EC_20_) concentration of 2-AG to screen for PAMs—compounds that enhance the response of the receptor to an agonist. In the assay design used here, it is possible that compounds may have activity on their own in the absence of 2-AG and would instead be classified as agonists. To distinguish between these possibilities, mycophenolate mofetil was tested for activity in the presence and absence of 2-AG in cells expressing rCB_2_/GIRK or rCB_1_/GIRK using thallium flux. Increasing concentrations of mycophenolate mofetil enhanced the response of rCB_2_/GIRK cells in a concentration-dependent manner, with similar potency and efficacy values in the absence of 2-AG (Figure 3, red circles, pEC_50_ = 6.81 ± 0.12, 154 nM; 71 ± 5% Max). Comparing this activity to the PAM assay performed in Figure 2A shows that the only difference in activity between the two experiments is the elevated baseline observed in the presence of an agonist due to the EC_20_ response of 2-AG. These results suggest that mycophenolate mofetil acts as an agonist as opposed to a PAM in rCB_2_/GIRK cells using thallium flux as a measure of activity. Mycophenolate mofetil decreased the response of rCB_1_/GIRK cells both in the presence (Figure 2A, blue circles) and absence (Figure 3, black circles, pEC_50_ < 5, −15 ± 3% Max) of 2-AG to a similar value, again differentiated only by the EC_20_ response of 2-AG. In rCB_1_/GIRK cells, mycophenolate mofetil decreases the signal below baseline when added alone, although weakly, which may suggest inverse agonist activity. The decrease in signal in the presence of 30 µM mycophenolate mofetil compared to baseline is significant when added alone (*p* = 0.002) and in the presence of 2-AG (*p* = 0.0003).

### 2.5. The Antagonist/Inverse Agonist AM630 Shifts the Mycophenolate Mofetil Concentration–Response Curve (CRC) to the Right Without Significantly Decreasing the Maximal Response

To further characterize the activity of mycophenolate mofetil, we investigated the effect of orthosteric antagonist/inverse agonist AM630 [37,38,39,40] on the concentration–response of rCB_2_ cells to mycophenolate mofetil. If AM630 and mycophenolate mofetil bind to the same site on the receptor, increasing concentrations of AM630 would be expected to shift the CRC to the right with no decrease in maximal response, a pattern indicative of competitive interaction. As seen in Figure 4, AM630 had little effect on the response to mycophenolate mofetil at or below a concentration of 1 µM. At 3 and 10 µM, AM630 significantly shifted the CRC to the right compared to vehicle (pEC_50_ vehicle = 6.47 ± 0.16; pEC_50_ 3 µM AM630 = 5.68 ± 0.09 (*p* = 0.01), pEC_50_ 10 µM AM630 = 5.64 ± 0.16 (*p* = 0.02)). There was no significant change in maximum signal compared to vehicle with either 3 or 10 µM AM630. While these results are not conclusive on their own, they suggest mycophenolate mofetil may have a competitive relationship with AM630.

### 2.6. Mycophenolate Mofetil Is Active in Phosphoinositide Hydrolysis Assays and Demonstrates Similar Activity in the Presence and Absence of 2-AG

While mycophenolate mofetil demonstrated potent agonist activity in thallium flux assays, it is important to confirm its activity using an assay that utilizes an alternate signaling pathway, as CB receptors exhibit assay-dependent pharmacology [23,26,27,41,42,43,44]. In a complementary assay, we used phosphoinositide (PI) hydrolysis in the presence of the chimeric G protein, G_qi9_, to assess activity. As shown in Figure 5, increasing concentrations of mycophenolate mofetil significantly increased PI hydrolysis in a concentration-dependent manner. Similar results were found in the presence (white circles, pIC_50_ = 6.15 ± 0.09, 700 nM; 633 ± 122 Max IP1) and absence (red circles, pIC_50_ = 6.28 ± 0.11, 521 nM; 438 ± 32 Max IP1) of 2-AG. These data are consistent with results from thallium flux assays, demonstrating that mycophenolate mofetil acts as a CB_2_ agonist in two distinct signaling pathways.

### 2.7. The CB_2_ Receptor Agonist GW-842,166X Exhibits a Similar Functional Activity Profile Compared to Mycophenolate Mofetil

GW-842,166X is a CB_2_ receptor agonist that demonstrates analgesic and anti-inflammatory activity in animal models [16,45,46] and has been evaluated in clinical trials for inflammation and pain [1,47]. It is structurally distinct from mycophenolate mofetil and was selected as a comparator for these studies (see Table 1 for structure). When tested for activity in rCB_2_/GIRK cells using thallium flux, GW-842,166X exhibited potent activity in the presence (white circles, pEC_50_ = 7.13 ± 0.08, 74 nM; 82 ± 4% Max) and absence (red circles, pEC_50_ = 7.26 ± 0.05, 56 nM; 89 ± 5% Max) of 2-AG, as shown in Figure 6A. GW-842,166X was inactive at rCB_1_ in both the presence (blue circles) and absence (black circles) of 2-AG. Overall, GW-842,166X exhibits a similar functional activity profile in thallium flux assays as seen with mycophenolate mofetil.

### 2.8. In Contrast to Mycophenolate Mofetil and GW-842,166X, the CB_2_ Receptor Agonist MDA7 Exhibits Inverse Agonist Activity at rCB_2_ in Thallium Flux Assays

MDA7/NTRX-07 is a CB_2_ receptor modulator structurally distinct from both mycophenolate mofetil and GW-842,166X, and has been described as exhibiting agonist activity at CB_2_ in GTPγS assays [18,19,20]. Studies have reported that MDA7 is active in models of inflammation and neuropathic pain [20,48,49,50], and the compound has been shown to induce neuroprotective effects in rodent models [51,52,53]. Currently, MDA7 is being evaluated as a novel therapeutic for the treatment of Alzheimer’s disease [1,47]. To compare it with the other agonists described above, we tested MDA7 for activity in both rCB_2_/GIRK and rCB_1_/GIRK cells using thallium flux. In contrast to both mycophenolate and GW-842,166X, MDA7 exhibited inhibitory activity in the presence (white circles, pEC_50_ < 5, −65 ± 15 Max) and absence (red circles, pEC_50_ < 5, −64 ± 13 Max) of 2-AG stimulation of rCB_2_, as shown in Figure 6B. In contrast, MDA7 activated rCB_1_ in the presence (blue circles, pEC_50_ = 6.40 ± 0.02, 395 nM; 48 ± 2% Max) and absence (black circles, pEC_50_ = 6.32 ± 0.03, 477 nM; 50 ± 2% Max) of 2-AG (Figure 6B). The detection of opposing CB_2_ activity in assays that target differential signaling pathways is difficult to predict, but is not unanticipated, as it has been observed with other CB receptor modulators [26,27,54,55,56,57,58,59].

### 2.9. Mycophenolate Mofetil and Other CB_2_ Modulators Differentially Bind to the Site Occupied by [^3^H]CP55,940

Results from functional assays, including thallium flux and PI hydrolysis, indicate that mycophenolate mofetil acts as an agonist of rCB_2_ due to its ability to activate the receptor when added alone to cells expressing rCB_2_; however, these assays do not provide information about the binding interaction between the receptor and mycophenolate mofetil. Agonists that bind to the receptor at the site occupied by the endogenous ligand (2-AG in this case) are considered orthosteric agonists. It is also possible for an agonist to bind and activate a receptor via a site other than the orthosteric site; these ligands are classified as allosteric agonists. To characterize the nature of the agonist activity of mycophenolate mofetil, we tested its ability to bind to the receptor in equilibrium binding assays using membranes harvested from cells expressing rCB_2_ using the radioligand [^3^H]CP55,940 [40,60]. CP55,940 has an affinity for both CB_1_ and CB_2_ receptors in the low nM range and is reported to bind to the site occupied by 2-AG [61]. The CB_2_ antagonist/inverse agonist AM630 fully inhibited radioligand binding with a pK_i_ value of 7.61 ± 0.29—results consistent with those described in the literature [40]. Increasing concentrations of mycophenolate mofetil had little effect on the equilibrium binding of [^3^H]CP55,940 until reaching 1 µM, and did not fully displace the binding when tested up to 30 µM (Figure 7 and Table 1). Similar results were observed for the agonist GW-842,166X. The incomplete binding and discrepancy between the functional potency and binding affinity of these ligands suggest an overlapping or alternate binding site as opposed to a competitive interaction. In contrast, the CB_2_ modulator MDA7 fully inhibited radioligand binding with a pK_i_ value of 6.60 ± 0.05. These binding results on their own suggest that MDA7 occupies the same site as [^3^H]CP55,940; however, the inverse agonist functional results suggest additional assays are needed to characterize the interaction.

### 2.10. Mycophenolate Mofetil, GW-842,166X, and MDA7 Behave as Agonists in cAMP and β-Arrestin Assays

Recent publications highlight the complex pharmacology of CB_2_ modulators such as EC21a and the importance of profiling CB_2_ modulators in assays that target different signaling pathways [26,27,43,55,59]. We have described results from thallium flux assays, which measure CB receptor-mediated opening of GIRK channels, and PI hydrolysis assays measuring activity via a chimeric G protein. We also evaluated the ability of mycophenolate mofetil and its comparators to modulate CB_2_ through two additional pathways: cAMP inhibition and β-arrestin recruitment. When tested in a cAMP inhibition assay (Figure 8A), in the presence or absence of an EC_20_ concentration of the agonist CP55,940, mycophenolate mofetil, GW-842,166X, and MDA7 all demonstrated agonist activity in cells expressing human CB_2_ receptors. The potency values for each (Table 2) were right-shifted compared to those observed in the thallium flux assay (Table 1). Mycophenolate mofetil also exhibited lower efficacy in cAMP assays compared to thallium flux, as well as versus GW-842,166X and MDA7. Similarly, when tested in a β-arrestin recruitment assay (Figure 8B) in cells expressing rat CB_2_ receptors, mycophenolate mofetil, GW-842,166X, and MDA7 also induced similar responses in the presence or absence of CP55,940. Potency values were generally right-shifted compared to those in thallium flux assays. MDA7 was notable in that its efficacy was lower than the other two ligands, both with and without 2-AG, when β-arrestin recruitment was measured. These data confirm that mycophenolate mofetil and the comparator GW-842,166X exhibit agonist activity across multiple assays and signaling pathways, and also confirm the signal bias induced by MDA7.

## 3. Discussion

The endocannabinoid system, composed of CB_1_ and CB_2_ receptors, has been extensively studied as a possible therapeutic target for various neurological and inflammatory disorders. CB_1_ is widely expressed across brain regions, while CB_2_ is primarily expressed in the immune system and microglia [1,2]. Recent studies have shown CB_2_ expression in hippocampal, striatal, and dopaminergic neurons, suggesting the potential for CB_2_ modulation for the treatment of disorders including pain, addiction, and schizophrenia [1,13,14,15,47,62,63]. Reduced CB_2_ receptor expression and activity are correlated with an increased risk of developing schizophrenia [64,65,66,67]. We have shown that the antipsychotic effects of muscarinic M_4_ receptor or metabotropic glutamate receptor 1 (mGlu_1_) PAMs in rodent behavioral models are dependent on CB_2_ activation, which occurs via retrograde signaling of 2-AG [14,15]. For these reasons, it is possible that CB_2_ activation or potentiation may represent a novel strategy for treatment. The use of a PAM as opposed to an agonist may provide advantages, as a PAM would be predicted to produce lower levels of receptor desensitization and greater receptor specificity [22,23]. We recently profiled the first published small molecule PAM, EC21a, showing that this ligand exhibits complex pharmacology [26]. To discover novel CB_2_ PAMs with improved properties compared to EC21a, we conducted a screen of FDA-approved drugs, resulting in the discovery of immunosuppressant mycophenolate mofetil as a selective activator of CB_2_.

The initial design of the assay was intended to screen for PAMs, but subsequent assays determined mycophenolate mofetil was active in the absence of an agonist. Other CB_2_ agonists have been or are currently being studied in preclinical and clinical trials, including GW-842,166X for pain [16,45,46] and MDA7/NTRX-07 for pain [49,50,68] and Alzheimer’s disease [51,52], but to date, no CB_2_ agonist drugs are approved for clinical use. As shown here, both mycophenolate mofetil and GW-842,166X present clear profiles as selective agonists of CB_2_ in multiple assays assessing various signaling pathways, including thallium flux, cAMP, and β-arrestin assays. MDA7 instead exhibits a mixed profile. In cAMP and β-arrestin assays, it acts as an agonist of the CB_2_ receptor; however, in thallium flux assays, MDA7 demonstrates inverse agonist activity, clearly blocking the basal signal well below baseline levels. It is also not selective for CB_2_ in thallium flux assays as it also significantly activates the CB_1_ receptor.

The binding data generated with [^3^H]CP55,940 indicate multiple binding modes/sites are present within the CB_2_ receptor. 2-AG is reported to share the same orthosteric binding site as CP55,940 [61], and we confirmed that 2-AG fully displaced radioligand binding (pK_i_ = 5.46 ± 0.05). Additionally, the inverse agonist AM630 fully inhibited the equilibrium binding of the radioligand with a K_i_ value consistent with that reported in the literature (pK_i_ = 7.61 ± 0.29) [40]. Mycophenolate mofetil was only able to inhibit radioligand binding to a level equal to 20% of [^3^H]CP55,940 total binding at 30 µM. This level of binding indicates a potential interaction with the binding site occupied by the radioligand, but the disconnect between the binding data and the functional potency generated in the same cell line (rCB_2_/GIRK) using thallium flux suggests the binding of mycophenolate mofetil is not completely competitive with [^3^H]CP55,940. It is possible the two ligands occupy the same site but have different binding contacts or that the ligands share an overlapping portion of the site. It will be important to fully characterize the binding mode of mycophenolate mofetil using mutagenesis and additional molecular pharmacology studies to more fully understand the binding and functional relationship and how it relates to future drug development. This is also the case for GW-842,166X, which exhibited a similar binding profile to that of mycophenolate mofetil with incomplete binding reaching 20% of [^3^H]CP55,940 total binding. The functional potency of this compound in thallium flux assays revealed a left-shifted pEC_50_ compared to the K_i_. MDA7, on the other hand, presents a different profile and fully inhibits the equilibrium binding of [^3^H]CP55,940 (pK_i_ = 6.60 ± 0.05). We have not tested the effect of AM630 on the activity of MDA7 or GW-842,166X in our laboratory; however, the effects of GW-842,166X in a preclinical model of inflammatory pain were reversed by AM630, suggesting the antihyperalgesia properties of GW-842,166X are mediated by CB_2_ [16]. AM630 was also found to reverse the antiallodynic effect of MDA7 in two rat models of neuropathic nociception [20]. In the rCB_2_/GIRK cell line measuring thallium flux, MDA7 displayed weak inverse agonist activity, with a potency value significantly right-shifted in comparison (pEC_50_ < 5.0). The complete inhibition observed in the binding assay suggests competitive interaction with the [^3^H]CP55,940 binding site, but the weak blockade in the functional assay indicates a disconnect between binding affinity and functional potency in the modulation of GIRK1/2 channels. MDA7 was initially reported as a CB_2_ receptor agonist in GTPγS assays [18,19,20], and we confirmed agonist activity in both cAMP inhibition and β-arrestin recruitment assays. We report herein that MDA7 acts as an inverse agonist when tested for activity using thallium flux. We observed a similar profile for the CB_2_ modulator EC21a, which was originally described as a PAM in a GTPγS assay; we confirmed PAM activity in cAMP inhibition, but EC21a also exhibited inverse agonist activity in CB_2_ thallium flux assays [26]. Binding data suggested EC21a could act as an inverse agonist via binding to an allosteric site on the CB_2_ receptor. MDA7 binding data generated using the CB_2_/GIRK cell line used in thallium flux assays showed complete inhibition of [^3^H]CP55,940, suggesting the two compounds occupy the same site. It is possible the MDA7 interaction is not fully competitive with the orthosteric site and instead involves a partial or overlapping interaction, which could explain the difference in pharmacology observed in this assay/cell line. Additional experiments, such as a Schild analysis or mutational studies, could be performed in the future to further investigate the binding mode of MDA7.

Mycophenolate mofetil is an ester prodrug of the active immunosuppressant mycophenolic acid and is rapidly hydrolyzed in vivo to liberate the parent compound. As mycophenolic acid was not found to exhibit CB_2_ activity, medicinal chemistry strategies to circumvent esterase-mediated hydrolysis will be needed for further development of this chemotype in the context of CB_2_. Toward this goal, we examined several well-established bioisosteric replacements for labile esters. Unfortunately, we observed that ester replacement with the direct secondary amide analog of mycophenolate mofetil resulted in a complete loss of CB_2_ activity. Additional preliminary modifications to generate substituted 1*H*-tetrazole and 1,2,4-oxadiazole analogs were similarly devoid of CB_2_ activity. Further structural refinements will therefore be needed for continued development of this chemotype; the synthesis of next-generation mycophenolate mofetil analogs is currently ongoing in our laboratories.

In this study, we discovered that mycophenolate mofetil is a selective agonist of CB_2_ through screening of an FDA-approved library of drugs. Its agonist activity was confirmed in multiple assays assessing activity through different signaling pathways, including thallium flux, PI hydrolysis, cAMP inhibition, and β-arrestin recruitment. The activity profiles of the CB_2_ clinical candidates GW-842,166X and MDA7 were assessed and found to be similar to mycophenolate mofetil in the case of GW-842,166X but divergent for MDA7. These findings suggest that modulators of CB_2_ have subtle pharmacological complexities, and understanding these differences may provide insight into developing better treatments for brain disorders such as schizophrenia, pain, and Alzheimer’s disease.

## 4. Materials and Methods

### 4.1. Chemicals

2-Arachidonoylglycerol (2-AG) and 6-iodo-2-methyl-1-[2-(4-morpholinyl)ethyl]-1*H*-indol-3-yl](4-methoxyphenyl)methanone (AM630) were purchased from Cayman Chemical (Ann Arbor, MI, USA). 2-[(1*R*,2*R*,5*R*)-5-hydroxy-2-(3-hydroxypropyl)cyclohexyl]-5-(2-methyloctan-2-yl)phenol (CP55,940) was purchased from Tocris Bioscience (Minneapolis, MN, USA). [^3^H]-CP55,940 was obtained from Perkin Elmer (Boston, MA, USA). 2-(2,4-dichloroanilino)-N-(oxan-4-ylmethyl)-4-(trifluoromethyl)pyrimidine-5-carboxamide (GW-842,166X) and 1-[(3-benzyl-3-methyl-2,3-dihydro-1-benzofuran-6-yl)carbonyl]piperidine (MDA7/NTRX-07) were synthesized by the Warren Center for Neuroscience Drug Discovery, Vanderbilt University, Nashville, TN, USA, as previously described [16,18,19,20].

### 4.2. Cell Culture

The cell lines used in this manuscript were generated in-house. Their preparation is described in detail in Qi et al. [26]. Cells were used for 10 passages for experiments and day-to-day signals in both the CB_2_ and CB_1_ lines were stable across this passage number. Human Embryonic Kidney (HEK) 293 cells stably expressing rCB_2_ and the G protein G_qi9_ were maintained in Dulbecco’s modified Eagle media (DMEM) containing 10% FBS, 1× antibiotic/antimycotic, 20 mM HEPES, 1 mM sodium pyruvate, 2 mM L-glutamine, 1× non-essential amino acids, 700 µg/mL G418 sulfate, and 0.6 µg/mL puromycin. HEK cells stably expressing rCB_2_ or rCB_1_ and the G protein inwardly rectifying potassium channel (GIRK) were maintained in DMEM/F12 containing 10% FBS, 1× antibiotic/antimycotic, 20 mM HEPES, 1 mM sodium pyruvate, 2 mM L-glutamine, 1× non-essential amino acids, 700 µg/mL G418 sulfate, and 0.6 µg/mL puromycin. Cells were monitored by periodic PCR detection using the LookOut Mycoplasma PCR Detection Kit (Sigma-Aldrich, St Louis, MO, USA) to eliminate potential mycoplasma infection. Reagents were obtained from Invitrogen (Waltham, MA, USA) unless otherwise noted.

### 4.3. Thallium Flux Assay

Thallium flux assays were performed as previously described in Niswender et al., 2008 [31]. HEK/GIRK cells stably expressing rCB_1_ or rCB_2_ (15,000 cells/20 µL/well) were seeded in 384-well, poly-D-lysine coated assay plates (Corning BioCoat^®^) (Corning Inc., Corning, NY, USA) and incubated overnight at 37 °C in the presence of 5% CO_2_. The next day, the medium was removed and replaced with 20 μL of 1.2 µM thallium-sensitive dye Thallos-AM (ION biosciences, San Marcos, TX, USA), prepared as a DMSO stock solution mixed in a 1:1 ratio with 10% (*w*/*v*) pluronic acid F-127, and diluted in assay buffer (Hank’s balanced salt solution, 20 mM HEPES, pH 7.4). Following 1 h at room temperature, the dye solution was removed and replaced with 20 µL assay buffer. For screening of the FDA-Approved Drug Collection, 10 mM DMSO stocks of library compounds were transferred to daughter plates using an Echo acoustic plate reformatter (Labcyte, Sunnyvale, CA, USA) and diluted in assay buffer to a 2× final concentration. For concentration–response curve experiments, compounds were serially diluted 1:3 into 10-point concentration–response curves in DMSO, transferred to daughter plates using the Echo or Bravo Automated Liquid Handling Platform (Agilent Technologies, Santa Clara, CA, USA), and diluted in assay buffer to a 2× final concentration. Thallium flux was measured using the Hamamatsu FDSS/μCELL Kinetic Plate Imager (Bridgewater, NJ, USA). After the establishment of a fluorescence baseline (excitation, 480 ± 20 nm; emission, 540 ± 30 nm), 20 µL (2×) of test compound was added to the cells at 1 s and the response was measured. After 140 s, 10 µL (5×) of an EC_20_ concentration of agonist (PAM mode) or vehicle (AGO mode) in thallium stimulus buffer (125 mM NaHCO_3_, 1.8 mM CaSO_4_, 1 mM MgSO_4_, 5 mM glucose, 12 mM Tl_2_SO_4_, 10 mM HEPES, 0.5% BSA, pH 7.4) was added to the cells, and the response of the cells was measured for an additional 159 s (data were acquired for 300 s total at 0.5 Hz for 140 s and 1 Hz for 160 s). Raw kinetic data were analyzed in a multi-step process: (1) Fluorescence readings for each time point in a well were divided by the fluorescence reading at the initial time point to account for differences in cell number, non-uniform illumination, and dye-loading. (2) The slope value for each kinetic trace was calculated for the time window of 145–155 s, a window occurring directly after the second addition. (3) The average slope was calculated for wells containing vehicle and this value was subtracted from all wells. (4) The vehicle-subtracted slope was normalized to the relevant maximal agonist signal for each assay. For concentration–response curves, normalized data were fit to a four-parameter logistic equation using GraphPad Prism 10 (La Jolla, CA, USA) or the Dotmatics software platform 6.2 (Dotmatics, Bishop’s Stortford, UK).$$y=bottom+\frac{top-bottom}{1+10^{\left(LogEC50-A\right)Hillslope}}$$
where *A* is the molar concentration of the compound; *bottom* and *top* denote the lower and upper plateaus of the concentration–response curve; Hillslope is the Hill coefficient that describes the steepness of the curve; and EC_50_ is the molar concentration of the compound required to generate a response halfway between the *top* and *bottom*. Data shown represent the mean ± standard error of the pEC_50_ or maximal response. Experiments were performed in duplicate or triplicate and repeated a minimum of three separate times.

### 4.4. PI Hydrolysis Assay

One day prior to experimentation, selected HEK/G_qi9_ monoclonal cells stably expressing rat CB_2_ were plated onto poly-D-lysine coated clear-bottom 384-well plates (15,000 cells per well) in DMEM supplemented with 10% FBS and 20 mM HEPES. Ten minutes prior to the assay, the cell culture medium was replaced with 20 µL 37 °C Hank’s balanced salt solution (HBSS; with Ca^2+^, Mg^2+^, glucose). IP_1_ stimulation was then initiated by adding 5 µL of 5× agonist dissolved in HBSS plus 40 mM Li^+^, and cells were incubated for an additional hour before aspiration and addition of Lysis Buffer. IP_1_ levels were determined using the Cisbio HTRF IP-ONE assay kit according to the manufacturer’s instructions, and fluorescence was measured using an Envision plate reader (PerkinElmer, Waltham, MA, USA). Data were acquired as HTRF ratio (665/620) and expressed as nanomolar levels of IP_1_.

### 4.5. Radioligand Binding Assays

Membranes were made from HEK/GIRK cells stably expressing rat CB_2_. Radioligand competition binding assays were performed as previously described, with minor modifications. In brief, compounds were serially diluted into assay buffer with 0.1% bovine serum albumin (BSA) and added to each well of a 96-well plate, along with 20 μg/well cell membrane and approximately 500 pM [^3^H]-CP55,940 (specific activity = 104 Ci/mmol, PerkinElmer, Waltham, MA, USA). Following a 3-h incubation period on a shaker at room temperature, membrane-bound ligand was separated from free ligand by filtration through glass fiber 96-well filter plates (Unifilter-96, GF/B; PerkinElmer, Waltham, MA, USA). Forty microliters of scintillation fluid were added to each well, and membrane-bound radioactivity was determined by scintillation counting (Microbeta2; Revvity, Waltham, MA, USA). Nonspecific binding was determined using 10 μM of cold CP55,940.

### 4.6. cAMP Assays

Assays were performed by Eurofins Discovery (Freemont, CA, USA). cAMP Hunter cell lines were expanded from freezer stocks according to standard procedures. Human CB_2_ cells were seeded in a total volume of 20 μL into white-walled, 384-well microplates and incubated at 37 °C for the appropriate time prior to testing. cAMP modulation was determined using the DiscoverX HitHunter cAMP XS+ assay. For agonist determination, cells were incubated with samples in the presence of EC_80_ forskolin to induce a response. Media was aspirated from cells and replaced with 10 μL HBSS/10 mM HEPES. Intermediate dilution of sample stocks was performed to generate 4× sample in assay buffer, and 5 μL of 4× sample was added to cells. Then, 5 μL of 4× EC_80_ forskolin diluted in HBSS/HEPES was added and incubated at 37 °C for 30 min. The final assay vehicle concentration was 1%. For allosteric determination, cells were pre-incubated with the sample followed by agonist induction at the EC_20_ concentration. Media was aspirated from cells and replaced with 10 μL HBSS/10 mM HEPES. Intermediate dilution of sample stocks was performed to generate 4× samples in assay buffer. Then, 5 μL of 4× compound was added to the cells and incubated at room temperature or 37 °C for 5 min. Subsequently, 5 μL of 4× EC_20_ agonist was added to the cells and incubated at 37 °C for 30 min. EC_80_ forskolin was included.

After appropriate compound incubation, the assay signal was generated by incubation with 5 μL cAMP XS+ Ab reagent, followed by 20 μL cAMP XS+ ED/CL lysis cocktail for one hour at room temperature. Then, 20 μL cAMP XS+ EA reagent was added and incubated for two hours at room temperature. Microplates were read following signal generation with a PerkinElmer Envision^TM^ instrument for chemiluminescent signal detection.

Compound activity was analyzed using the CBIS data analysis suite (ChemInnovation, San Diego, CA, USA). For agonist mode assays, percentage activity was calculated using the following formula: % Activity = 100% × (1 − (mean RLU of test sample − mean RLU of MAX control)/(mean RLU of vehicle control − mean RLU of MAX control)).

For Gi positive allosteric mode assays, percentage modulation was calculated using the following formula: % Modulation = 100% × (1 − (mean RLU of test sample − mean RLU of MAX control)/(mean RLU of EC_20_ control − mean RLU of MAX control)).

### 4.7. β-Arrestin Assays

Assays were performed by Eurofins Discovery (Freemont, CA, USA). PathHunter cell lines were expanded from freezer stocks according to standard procedures. Rat CB_2_ cells were seeded in a total volume of 20 μL into white-walled, 384-well microplates and incubated at 37 °C for the appropriate time prior to testing. For agonist determination, cells were incubated with the sample to induce a response. Intermediate dilution of sample stocks was performed to generate a 5× sample in assay buffer. A total of 5 μL of 5× sample was added to the cells and incubated at 37 °C or room temperature for at least 120 min. The final assay vehicle concentration was 1%. For allosteric modulator determination, cells were pre-incubated with the sample followed by agonist challenge at the EC_20_. Intermediate dilution of sample stocks was performed to generate a 5× sample in assay buffer. A total of 5 μL of 5× sample was added to cells and incubated at 37 °C or room temperature for 15 min. Vehicle concentration was 1%. Then, 5 μL of 6× EC_20_ agonist in assay buffer was added to the cells and incubated at 37 °C or room temperature for at least 120 min.

The assay signal was generated through a single addition of 15 μL (50% *v*/*v*) of PathHunter Detection reagent cocktail, followed by a one-hour incubation at room temperature. Microplates were read following signal generation with a PerkinElmer Envision^TM^ instrument for chemiluminescent signal detection.

Compound activity was analyzed using the CBIS data analysis suite (ChemInnovation). For agonist mode assays, percentage activity was calculated using the following formula: % Activity = 100% × (mean RLU of test sample − mean RLU of vehicle control)/(mean MAX control ligand − mean RLU of vehicle control).

For positive allosteric mode assays, percentage modulation was calculated using the following formula: % Modulation = 100% × ((mean RLU of test sample − mean RLU of EC_20_ control)/(mean RLU of MAX control ligand − mean RLU of EC_20_ control)).

## Figures and Tables

**Figure 1 ijms-26-04956-f001:**
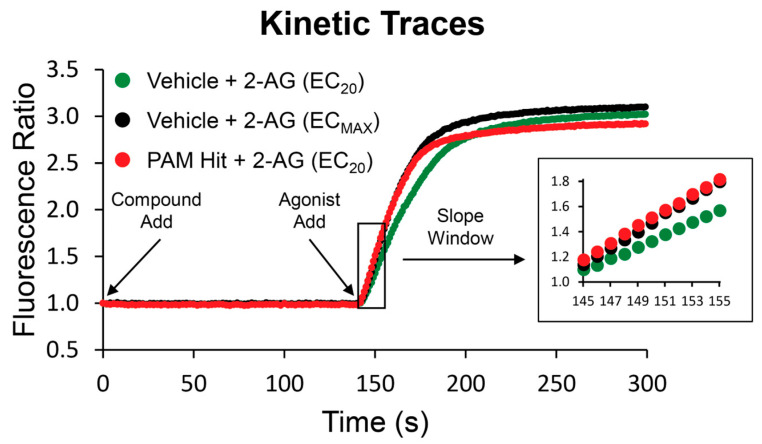
Sample traces from screen of FDA collection exemplify a PAM hit. Test compounds or vehicle were added at *t* = 1 s, and the fluorescence ratio was monitored. 2-AG (EC_20_ or EC_Max_) was added at *t* = 141 s. Fluorescence readings for each time point in a well were divided by the fluorescence reading at the initial time point to account for differences in cell number, non-uniform illumination, and dye-loading. The slope value for each kinetic trace was calculated for the time window of 145–155 s. The average slope was calculated for wells containing vehicle and this value was subtracted from all wells. Vehicle-subtracted slopes were normalized to the relevant maximal agonist signal for each assay. Traces for a sub-maximal agonist response (EC_20_, green) are compared to a maximal response (EC_Max_, black) and PAM hit (red).

**Figure 2 ijms-26-04956-f002:**
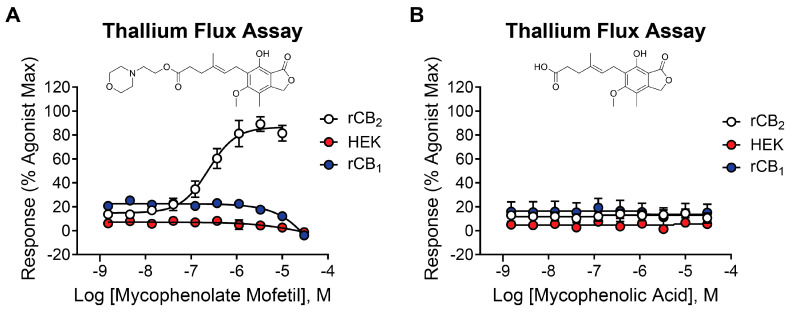
Mycophenolate mofetil is potent and selective for rCB_2_, while mycophenolic acid is inactive at CB receptors and parental HEK/GIRK cells. Concentration–response curves in thallium flux assays for (**A**) mycophenolate mofetil and (**B**) mycophenolic acid in the presence of a sub-maximal (EC_20_) concentration of agonist (2-AG or acetylcholine). Results are shown for rCB_2_ (white), HEK/GIRK (no CB receptor, red), and rCB_1_ (blue) cells. Data represent the mean ± SEM of at least three independent experiments run in duplicate or triplicate, except for mycophenolic acid at HEK/GIRK cells, which was run in two independent experiments. Data are plotted as a percentage of maximal 2-AG response for rCB_2_ and rCB_1_, or acetylcholine response for HEK/GIRK cells.

**Figure 3 ijms-26-04956-f003:**
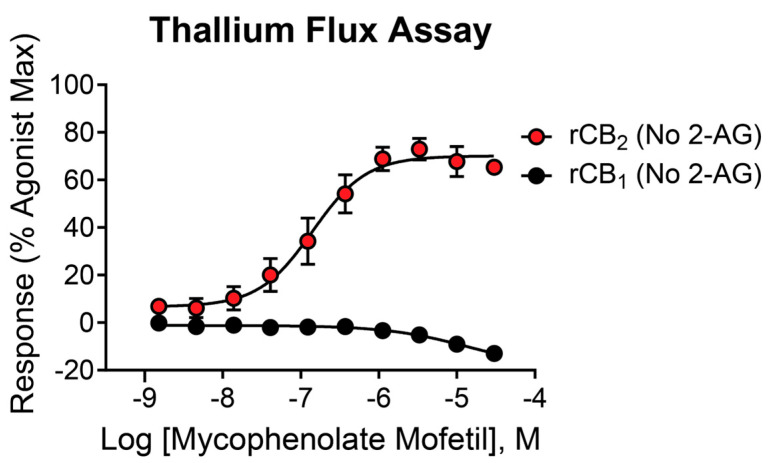
Mycophenolate mofetil is active at CB receptors in the absence of 2-AG. Concentration–response curves in thallium flux assays for mycophenolate mofetil in the absence of agonist. Results are shown for rCB_2_ (red) and rCB_1_ (black) cells. Data represent the mean ± SEM of at least three independent experiments run in duplicate or triplicate. Data are plotted as a percentage of maximal 2-AG response.

**Figure 4 ijms-26-04956-f004:**
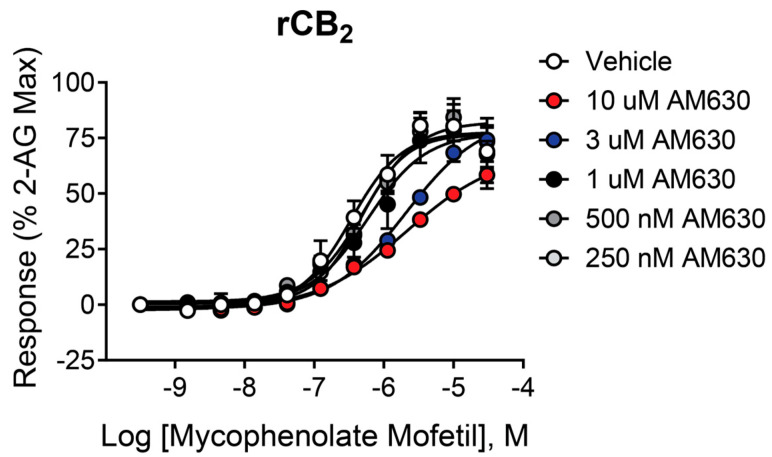
Increasing concentrations of AM630 shift the CRC of mycophenolate mofetil in cells expressing rCB_2_. Concentration–response curves (CRCs) in thallium flux assays for mycophenolate mofetil in the presence of AM630. Data represent the mean ± SEM of at least three independent experiments run in duplicate or triplicate. Data are plotted as a percentage of maximal 2-AG response.

**Figure 5 ijms-26-04956-f005:**
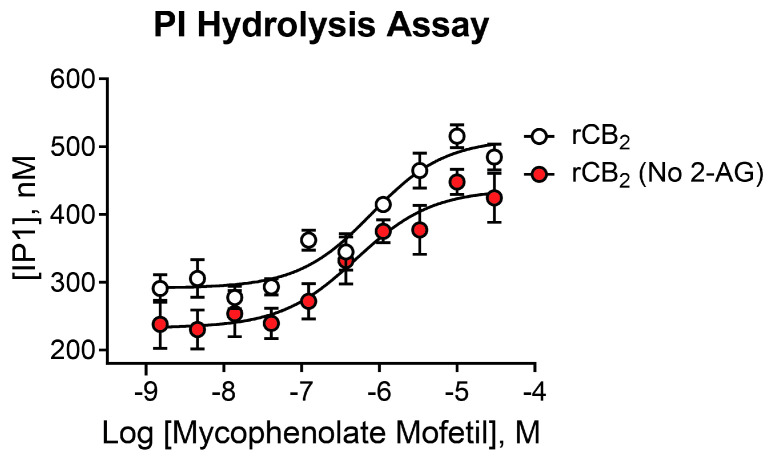
Mycophenolate mofetil exhibits similar activity in presence and absence of 2-AG in PI hydrolysis. Concentration–response curves in PI hydrolysis assays for mycophenolate mofetil in the presence (white) and absence (red) of a sub-maximal (EC_20_) concentration of agonist in cells expressing rCB_2_. Data represent the mean ± SEM of three independent experiments run in duplicate or triplicate. Data are plotted as IP1 concentration in nM.

**Figure 6 ijms-26-04956-f006:**
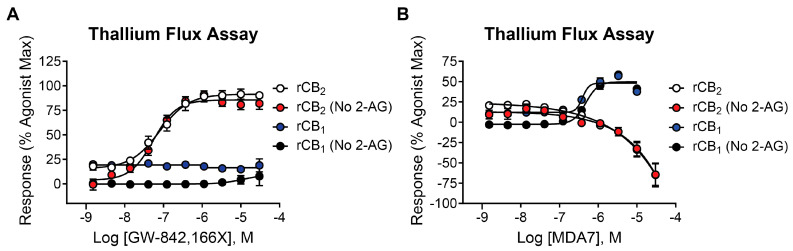
The CB_2_ agonists GW-842,166X and MDA7 exhibit distinctions in pharmacological activity in rCB_2_ thallium flux assays. Concentration–response curves in thallium flux assays are shown for GW-842,166X (**A**) and MDA7 (**B**). Results are presented for rCB_2_ in the presence (white) or absence (red) of a sub-maximal (EC_20_) concentration of agonist. Results are presented for rCB_1_ in the presence (blue) or absence (black) of a sub-maximal (EC_20_) concentration of agonist. Data represent the mean ± SEM of three independent experiments run in duplicate or triplicate. Data are plotted as a percentage of the maximal 2-AG response.

**Figure 7 ijms-26-04956-f007:**
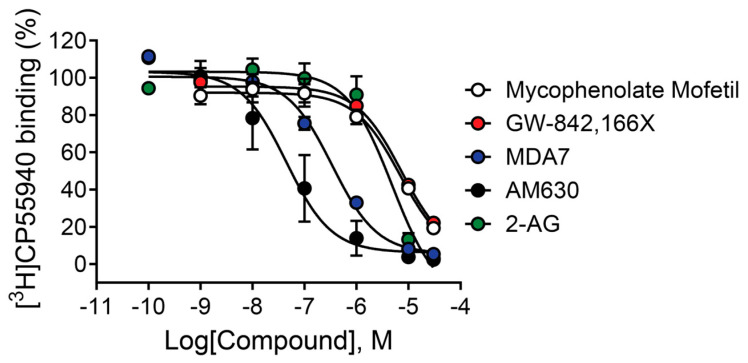
CB receptor ligands exhibit varying effects on [^3^H]CP55,940 binding under equilibrium conditions. Competition binding concentration–response curves were obtained in the presence of 0.5 nM [^3^H]CP55,940 using membranes harvested from HEK/GIRK cells expressing rCB_2_. Data represent the mean ± SEM of three independent experiments run in triplicate. Data are plotted as a percentage of specific [^3^H]CP55,940 binding. Non-specific binding was determined in the presence of 10 µM CP55,940.

**Figure 8 ijms-26-04956-f008:**
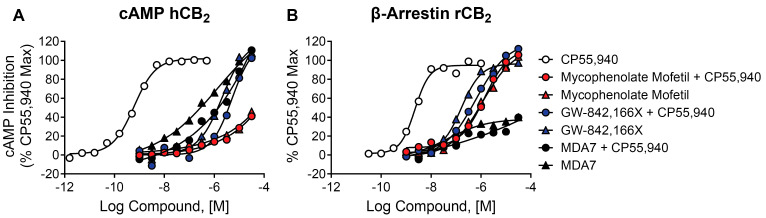
Differential activity of CB_2_ agonists in cAMP and β-arrestin assays. Concentration–response curves in (**A**) cAMP assays and (**B**) β-arrestin recruitment assays, in the presence or absence of an EC_20_ concentration of agonist CP55,940, are shown for mycophenolate mofetil (red), GW-842,166X (blue), and MDA7 (black). Control concentration–response curves for CP55,940 are shown in white. Data represent at least one independent experiment run in duplicate. Data are plotted as a percentage of the maximal CP55,940 response.

**Table 1 ijms-26-04956-t001:** Activity of CB_2_ compounds in thallium flux and binding assays.

Compound	rCB_2_ GIRK		rCB_1_ GIRK		HEK GIRK	rCB_2_ Binding
	(+) 2-AG pEC_50_ ± SEM % Max ± SEM	(−) 2-AG pEC_50_ ± SEM % Max ± SEM	(+) 2-AG pEC_50_ ± SEM % Max ± SEM	(−) 2-AG pEC_50_ ± SEM % Max ± SEM	(+) ACh pEC_50_ ± SEM % Max ± SEM	Binding Affinity pK_i_ ± SEM % Max ± SEM
Mycophenolate mofetil 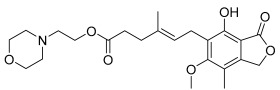	6.64 ± 0.06 74 ± 2%	6.81 ± 0.12 71 ± 5%	<5.0 −5.5 ± 1.1%	<5.0 −15 ± 3%	Inactive	Incomplete pEC_50_ < 5.0 20 ± 1
Mycophenolic acid 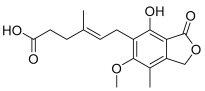	Inactive	ND	Inactive	ND	Inactive	ND
GW-842,166X 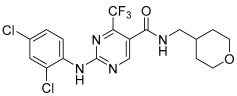	7.13 ± 0.08 82 ± 4%	7.26 ± 0.05 89 ± 5%	Inactive	Inactive	ND	Incomplete pEC_50_ < 5.0 22 ± 2
MDA7 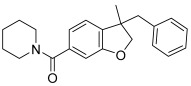	<5.0 −65 ± 15%	<5.0 −64 ± 13%	6.40 ± 0.02 48 ± 2	6.32 ± 0.03 50 ± 2	ND	6.60 ± 0.05 5.3 ± 1.7

ND indicates not determined; (+) or (−) 2-AG or ACh indicates the presence or absence of an EC_20_ concentration of agonist 2-AG or acetylcholine (ACh).

**Table 2 ijms-26-04956-t002:** Activity of CB_2_ compounds in cAMP and β-arrestin assays.

Compound	hCB_2_ cAMP		rCB_2_ β-Arrestin	
	(+) CP55,940 pEC_50_ % Max	(−) CP55,940 pEC_50_ % Max	(+) CP55,940 pEC_50_ % Max	(−) CP55,940 pEC_50_ % Max
Mycophenolate mofetil	<5.0 41	<5.0 46	5.87 113	5.94 114
GW-842,166X	5.29 119	5.69 120	6.20 120	6.79 97
MDA7	<5.0 111	<5.0 108	<5.0 40	7.32 38

(+) or (−) CP55,940 indicates the presence or absence of an EC_20_ concentration of agonist CP55,940.

## Data Availability

The authors declare that all processed data supporting the findings of this study are available within the paper. Raw data are available on request from the corresponding author.

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
