# Peer review of "Characterization of Novel and Known Activators of Cannabinoid Receptor Subtype 2 Reveals Mixed Pharmacology That Differentiates Mycophenolate Mofetil and GW-842,166X from MDA7"

_ijms, 2025, doi:10.3390/ijms26104956_

Round 1
Reviewer 1 Report
Comments and Suggestions for Authors
The most innovative aspects of the experiments presented in this manuscript lie in their methodological “sophistication” (GIRK/thallium assay, multi-pathway analysis) and, in my opinion, quite a “revelation” of the presented in vitro CB2 pharmacology in studied clinically relevant compounds.
Based on the obtained results, I think, identifying mycophenolate mofetil as a novel, selective CB2 agonist with non-orthosteric binding behavior may drive further studies towards non-classical (“novel” old) CB2 ligands and CB2 neurotransmission.
Extensively written methodology and resultant description draw the reader into scientific considerations.
The following are only minor comments of a rather technical nature.
- Materials and Methods
4.4. PI Hydrolysis Assay
It would be helpful to mention how the aforementioned HEK/Gqi9 monoclonal cell line stably expressing rat CB2 (also mentioned earlier about the CB1 variant) was selected. This may be in this subsection or the preceding, separate one. How was the stability of expression of the mentioned receptors evaluated? Briefly.
4.6. cAMP Assays
It would be helpful to mention - to explain, highlight the abbreviation RLU and the reasonableness of including the parameter EC20 as submaximal concentration (EC20) (mentioned in line 87) - for clarity of reading by the reader. This is only a suggestion for consideration, though.
- Results
2.1. Identification of novel modulators of the CB2 receptor
Line 93-94 - How about a list of 13 putative rCB2 modulators? Only a suggestion for consideration.
Reviewer 2 Report
Comments and Suggestions for Authors
The authors describe a quite comprehensive study of the pharmacology of 3 ligands of CB2 receptors. In light of the high potential of CB2 receptors for the treatment of different diseases, this work is important and interesting. The experiments are well-performed and well-presented, with only a few minor corrections:
- In kinetic traces, the graph should also show 2-AG alone for the comparison, without additional compounds. Or are the add-on compounds only presented by the red curve? It is unclear. As for the slope, it should probably be normalized also for 2-AG alone.
- In Table 1. As far as can be understood from the text, the activity of mycophenolate mofetil is independent, and it binds to the same binding site as 2AG. Then, why in the Table is the column headed as "PAM activity"? It is not PAM, but orthosteric activity.
- Does AM630 interfere with the effects of MPA7 and GW-842,166X?
- AM-630 binds to the same site as CP-55,940, so why does the binding assay with tritiated CP not show positive results? It has to be addressed in the discussion.
- In-vivo, mycophenolate mofetil will be very fast metabolized to the acis, and thus be inactive for CB2 activation. The possible ways to overcome it should be discussed.
- If mycophenolate mofetil binds to the same binding site as 2-AG, how are their activation extent when compared on the same graph?
- It seems that mycophenolate mofetil is a partial agonist for CB2, and thus 2-AG can add 20% to CB2 response, while GW-842,166X is a full agonist, and thus 2-AG addition does not change CB2 activation by it.
- Judging by Fig. 8, the compounds work quite the same with and without CP-55,940, it is better to position the curves on the same graphs to better see it.
- The statement that the compounds "enhanced the cAMP response to C55,940" cannot be done, as they have an independent high effect, while PAM cannot work by themselves, but only enhance the effect of orthosteric ligands.
- The controversial effect of MDA7 has to be deeply discussed. How comes it seems to act as an agonist in cAMP assay, while acting as an antagonist in other assays?
- In methods, the transfected cells have to be better described. If they are commercial, the cal. number and the company have to be shown. If they are prepared in the authors' lab, the preparation details should be presented.
